# Impact of csDMARDs vs. b/tsDMARDs on the Prognosis of Rheumatoid Arthritis-Associated Interstitial Lung Disease: A Multicenter, Retrospective Study

**DOI:** 10.3390/diagnostics15070800

**Published:** 2025-03-21

**Authors:** Kyung-Ann Lee, Bo Young Kim, Sung Soo Kim, Yun Hong Cheon, Sang-Hyon Kim, Jae Hyun Jung, Geun-Tae Kim, Jin-Wuk Hur, Myeung-Su Lee, Chong Hyuk Chung, Yun Sung Kim, Seung-Jae Hong, Hae-Rim Kim, Hong Ki Min, Se Hee Kim, Su-Jin Moon, Sung Hae Chang, Soojin Im, Bo Da Nam, Hyun-Sook Kim

**Affiliations:** 1Division of Rheumatology, Department of Internal Medicine, Soonchunhyang University Seoul Hospital, Seoul 04401, Republic of Korea; cyberag@naver.com; 2Division of Rheumatology, Department of Internal Medicine, Gangneung Asan Hospital, Ulsan University College of Medicine, Gangneung 25440, Republic of Korea; 44113795by@gmail.com (B.Y.K.); drkiss@ulsan.ac.kr (S.S.K.); 3Department of Rheumatology, Gyeongsang National University School of Medicine and Gyeongsang National University Hospital, Jinju 52727, Republic of Korea; hong369c@naver.com; 4Institute of Health Sciences, Gyeongsang National University, Jinju 52828, Republic of Korea; 5Division of Rheumatology, Department of Internal Medicine, Keimyung University Dongsan Medical Center, Daegu 42601, Republic of Korea; mdkim9111@hanmail.net; 6Division of Rheumatology, Department of Internal Medicine, Korea University Ansan Hospital, Ansan 15355, Republic of Korea; cjhtmod@korea.ac.kr; 7Division of Rheumatology, Department of Internal Medicine, Kosin University Gospel Hospital, Busan 49267, Republic of Korea; gtah@hanmail.net; 8Division of Rheumatology, Department of Internal Medicine, Nowon Eulji Medical Center, Seoul 01830, Republic of Korea; mdjwhur@hanmail.net; 9Division of Rheumatology, Department of Internal Medicine, Wonkwang University Hospital, Iksan 54538, Republic of Korea; ckhlms@hanmail.net (M.-S.L.); taylorchung@hanmail.net (C.H.C.); 10Division of Rheumatology, Department of Internal Medicine, Chosun University Hospital, Gwangju 61453, Republic of Korea; denie117@chosun.ac.kr; 11Division of Rheumatology, Department of Internal Medicine, Kyung Hee University Medical Center, Seoul 02447, Republic of Korea; hsj718@hanmail.net; 12Division of Rheumatology, Department of Internal Medicine, Research Institute of Medical Science, Konkuk University Medical Center, Konkuk University School of Medicine, Seoul 05030, Republic of Korea; 20050039@kuh.ac.kr (H.-R.K.); 20190194@kuh.ac.kr (H.K.M.); 13Division of Rheumatology, Department of Internal Medicine, Kyung Hee University College of Medicine, Kyung Hee University Hospital at Gangdong, Seoul 05278, Republic of Korea; 02230002@khnmc.or.kr; 14Division of Rheumatology, Department of Internal Medicine, Yeouido St. Mary’s Hospital, College of Medicine, The Catholic University of Korea, Seoul 07345, Republic of Korea; prajna79@catholic.ac.kr; 15Division of Rheumatology, Department of Internal Medicine, Soonchunhyang University College of Medicine, Cheonan 31151, Republic of Korea; cocojasmin.st.grace@gmail.com; 16RexSoft Inc., Seoul 08826, Republic of Korea; soojin@rexsw.com; 17Departments of Public Health Sciences, Seoul National University, Seoul 08826, Republic of Korea; 18Department of Radiology, Soonchunhyang University Seoul Hospital, Soonchunhyang University School of Medicine, Seoul 04401, Republic of Korea; namboda@naver.com; 19Department of Radiology, Chung-Ang University Hospital, Chung-Ang University School of Medicine, Seoul 06974, Republic of Korea

**Keywords:** arthritis, rheumatoid, lung diseases, interstitial, prognosis, disease progression, antirheumatic agents

## Abstract

**Background/Objectives**: Rheumatoid arthritis-associated interstitial lung disease (RA-ILD) significantly affects disease prognosis and patient survival. The impact of conventional synthetic DMARDs (csDMARDs) and biologic/targeted synthetic DMARDs (b/tsDMARDs) on RA-ILD prognoses remains unclear. This study aimed to investigate the effects of csDMARDs and b/tsDMARDs on RA-ILD progression and prognosis based on pulmonary function tests (PFTs), high-resolution computed tomography (HRCT), and symptom changes. **Methods**: This multicenter, retrospective, observational study included patients with RA-ILD at 13 referral hospitals in South Korea. The participants were categorized into csDMARD-only and b/tsDMARD-exposed groups. RA-ILD prognosis was assessed over a 24-month follow-up period using serial PFTs (the forced vital capacity [FVC] and diffusing capacity of the lungs for carbon monoxide [DLCO]), HRCT findings, and clinical symptom changes. Kaplan–Meier survival analyses and Cox proportional hazards models were used to compare disease progression risk while adjusting for baseline lung function, RA disease activity, and glucocorticoid use. **Results**: Among 127 eligible patients, 22 (17.3%) were exposed to b/tsDMARDs, predominantly abatacept and tocilizumab. During a mean follow-up of 2.8 years, 65 (51.2%) patients experienced RA-ILD progression. A higher baseline Disease Activity Score-28 with erythrocyte sedimentation rate (DAS28-ESR) (adjusted hazard ratio [aHR]: 1.344, 95% confidence interval [CI]: 1.136–1.590, *p* = 0.001) and initially prescribed prednisone dose (aHR: 1.078, 95% CI: 1.011–1.151, *p* = 0.023) were significant prognostic factors for ILD progression. No statistically significant difference in progression risk was observed between the csDMARD-only and b/tsDMARD-exposed groups (aHR: 0.937, *p* = 0.851). **Conclusions**: The RA-ILD prognosis was more strongly influenced by disease activity, rather than the type of DMARD used. These findings emphasize the importance of maintaining low RA disease activity to improve RA-ILD prognosis.

## 1. Introduction

Rheumatoid arthritis (RA) is a chronic autoimmune disease characterized by progressive joint destruction, and it is often accompanied by interstitial lung disease (ILD), a complication that substantially contributes to morbidity and mortality [1]. The course of RA-associated ILD (RA-ILD) widely varies among patients, with minimal or slow disease progression being observed in some patients and more rapid deterioration occurring in others [2].

Older age, a reduced forced vital capacity (FVC) or diffusing capacity of the lungs for carbon monoxide (DLCO), a greater extent of ILD, and the presence of a usual interstitial pneumonia (UIP) pattern on high-resolution computed tomography (HRCT) have been identified through several studies as risk factors for mortality in patients with RA-ILD [3,4]. Fewer studies have investigated risk factors for lung-function deterioration, and the risk factors associated with a faster decline in lung function appear to be the same as those associated with mortality. Additionally, growing evidence suggests that higher RA disease activity is associated with or may contribute to RA-ILD progression [5,6]. Nevertheless, a prospective study involving 140 patients found no significant association between the trajectory of Disease Activity Score-28 with erythrocyte sedimentation rate (DAS28-ESR) and the trajectory of lung function [7].

The management of RA-ILD poses considerable challenges, owing to concerns about the potential lung toxicity of disease-modifying antirheumatic drugs (DMARDs) [1,2], which are widely used for managing RA. Furthermore, the potential benefits of DMARDs in RA-ILD treatment remain uncertain, and the use of conventional synthetic DMARDs (csDMARDs) (e.g., methotrexate and leflunomide) and biologic/targeted synthetic DMARDs (b/tsDMARDs) (e.g., tumor necrosis factor [TNF] inhibitors) in RA-ILD treatment is surrounded with controversy. Some studies have demonstrated improvement or stabilization, whereas others have reported RA-ILD development or progression [8,9,10]. Accumulative evidence suggests that non-TNF inhibitors (e.g., abatacept, tocilizumab, and rituximab) can reduce the risk of worsening RA-ILD [1,11,12,13]. Nonetheless, studies comparing the impact of b/tsDMARDs on RA-ILD progression to that of csDMARDs are lacking. Therefore, this multicenter, retrospective study aimed to assess the impact of csDMARDs and b/tsDMARDs on RA-ILD progression and prognosis.

## 2. Materials and Methods

### 2.1. Study Design and Population

This multicenter, retrospective, observational study enrolled patients with RA-ILD who met the 2010 American College of Rheumatology (ACR)/European League Against Rheumatism classification criteria or 1987 ACR classification criteria for RA at 13 referral hospitals across South Korea [14,15]. RA-ILD was confirmed via HRCT [16]. The inclusion criteria were as follows: (1) DMARD treatment for at least 3 months; (2) age between 19 and 75 years at the time of the first DMARD treatment; and (3) patients with RA-ILD who underwent at least two pulmonary function tests (PFTs) and/or HRCT scans within 6 months before or after the first DMARD treatment and then after a follow-up interval of at least 6 months. Patients with other connective tissue diseases (except for secondary Sjögren’s syndrome), incomplete clinical data, or a history of malignancy/radiation therapy were excluded.

### 2.2. Ethics Approval

This analysis was conducted in accordance with the ethical principles embodied in the Declaration of Helsinki. The study protocol and data collection forms were approved by the institutional review boards (IRBs) or local ethics committees of all participating centers (IRB number SCH 2021-07-011), which waived the requirement for the acquisition of informed consent from patients, owing to the retrospective nature of this study and to the use of anonymized clinical data in the analysis.

### 2.3. Definitions of Outcomes

The primary outcome was ILD progression during treatment with csDMARDs or b/tsDMARDs. RA-ILD progression was assessed by conducting follow-up examinations (including PFTs, DLCO, and HRCT) according to a clinical expert’s judgment.

Progression was defined as meeting any of the following criteria within 24 months: (1) a relative decline in the FVC of at least 10% of the predicted value; (2) a relative decline in the FVC of 5% to less than 10% of the predicted value, accompanied by worsening respiratory symptoms or an increased extent of fibrosis in HRCT; or (3) worsening respiratory symptoms and increased extent of fibrosis [17]. Accordingly, patients without these findings were deemed to have non-progressive RA-ILD.

Regarding chest HRCT scans, the increased extent of fibrosis was qualitatively assessed by a board-certified thoracic radiologist at each participating center by comparing it with the baseline HRCT. The assessment was primarily based on the extent and phenotype of ILD. ILD was classified into nonspecific interstitial pneumonia, UIP including probable UIP, or other patterns, including bronchiolitis obliterans, organizing pneumonia, lymphocytic interstitial pneumonitis, and mixed patterns. The presence of a honeycombing pattern in HRCT was also recorded [18]. The predicted values for FVC and DLCO were derived using Korean reference standards [19,20]. Worsening respiratory symptoms were defined as either worsening dyspnea or the onset of a new cough. Worsening dyspnea was assessed using the modified Medical Research Council scale (0–4) and was defined as an increase of at least one point.

Patients’ clinical data, including the index date, were extracted from the hospital’s electronic medical records. For outcome evaluation during DMARD administration, the index date for study participants was defined as the date of their first DMARD prescription. Each participant was followed until the earliest occurrence of one of the following events: ILD progression, drug discontinuation, death, or the last observation date. Drug discontinuation was defined as the absence of DMARD prescription records for a period exceeding the usual drug administration interval plus 91 days.

### 2.4. Main Exposure and Covariates

The main exposure was the csDMARD and b/tsDMARD prescription status. Patients were divided into two groups according to the type of medication that they received—namely, (1) the csDMARD-only group, which included patients who were exclusively treated with csDMARDs during the observation period, and (2) the b/tsDMARD-exposed group, which comprised patients who received b/tsDMARDs for at least 3 months during the observation period. If the interval between b/tsDMARD treatment initiation and the endpoint was less than 3 months, the observation period was limited to the time prior to b/tsDMARD treatment initiation, and patients were assigned to the csDMARD-only group. If the patients started with csDMARDs and later added b/tsDMARDs, this was treated as a time-varying exposure, with the switch point defined as the time of bDMARD initiation. The observation period was considered until the last recorded b/tsDMARD prescription. If the patients started with b/tsDMARDs and later added csDMARDs, these patients were classified as b/tsDMARD users from the baseline. The observation period was considered until the last recorded b/tsDMARD prescription.

Potential confounders considered in the analysis were age, sex, smoking status (current or ever-smoker), prednisone dose, rheumatoid factor positivity, anti-cyclic citrullinated peptide positivity, C-reactive protein (CRP) levels, ESR, DAS28-ESR [21], percentage predicted FVC, percent predicted DLCO, and honeycombing pattern in HRCT. These variables were included as covariates in our analyses. For ever-smoking, a person must have reported smoking at least 100 cigarettes in their lifetime, while a current smoker was someone who had met this criterion and continued to smoke cigarettes.

### 2.5. Statistical Analysis

Baseline characteristics were compared according to progression among patients with RA-ILD using the t-test for continuous variables and the chi-squared test for categorical variables. If >20% of cells had an expected frequency of ≤5, the chi-squared test was replaced with Fisher’s exact test. Survival probabilities for ILD progression in the csDMARD-only and b/tsDMARD-exposed groups were estimated using the Kaplan–Meier method, and the statistical significance of differences between the groups was assessed using the log-rank test. Confidence intervals (CIs) for survival curves were determined using Greenwood’s formula.

Hazard ratios (HRs) for the b/tsDMARD-exposed group during the observation period were calculated using stratified time-dependent Cox proportional hazards models. Stratification by index year (2010s or 2020s) was performed to adjust for measurement bias because more recent outcome assessment data were collected at shorter intervals compared to earlier data. The b/tsDMARD-exposed group included individuals who were treated exclusively with b/tsDMARDs, those who switched from csDMARDs to b/tsDMARDs, and those who were treated with the combination of b/tsDMARDs and csDMARDs. To reflect changes in drug exposure over time, the type of medication was considered a time-varying covariate based on the start time of b/tsDMARD exposure; this approach aids in preventing bias and improves the statistical power by utilizing all follow-up data. CRP values were log-transformed to achieve a normal distribution before statistical analysis. The original CRP data exhibited a skewness of 2.93, indicating substantial deviation from normality (skewness > 1). To mitigate this skewness and approximate a normal distribution, a natural logarithm transformation was applied. After log transformation, the skewness was reduced to −0.12, ensuring a more symmetric distribution suitable for parametric analysis.

Stepwise selection with the Akaike information criterion (AIC) was performed to select and adjust for known relevant variables. Missing values for covariates in the Cox proportional hazards models were imputed using the mean and mode when the missing rate was below 15% or the missForest algorithm when the missing rate exceeded 15%. Variables imputed using the missForest algorithm were the percentage of predicted FVC and the percentage of predicted DLCO, with acceptable imputation performance indicated by a normalized root mean square error of 0.23.

Statistical analyses were performed using R version 4.2.3 (R Foundation for Statistical Computing, Vienna, Austria) and Rex version 3.0.3 (RexSoft Inc., Seoul, Republic of Korea), Excel-based statistical software, with statistical significance set at a two-sided level of 0.05.

## 3. Results

### 3.1. Patient Characteristics

Among the recruited patients with RA-ILD, 127 individuals were included in the statistical analyses based on the inclusion and exclusion criteria, with 22 (17.3%) patients exposed to b/tsDMARDs during the follow-up period. For these 22 patients, abatacept was the most commonly used agent (*n* = 13), followed by tocilizumab (*n* = 7). Among the patients exposed to b/tsDMARDs, 17 used both csDMARDs and b/tsDMARDs concurrently. Of these, 14 patients initially started with csDMARDs and later added b/tsDMARDs, while 3 patients initially started with b/tsDMARDs and later added csDMARDs. Among the concomitantly administered csDMARDs, MTX was the most commonly used, with 11 patients. During the mean (standard deviation [SD]) follow-up period of 1058.05 (907.62) days, ILD progression occurred in 65 patients (51.2%). The patients’ baseline and follow-up characteristics are summarized in Table 1, with imputed variables being presented along with the number of missing values and their distribution after imputation.

The mean (SD) age at the baseline was 63.0 (8.2) years, with male patients accounting for 44.88% (*n* = 57). The mean (SD) FVC and DLCO at the baseline were 89.17% (17.14) and 69.55% (16.91), respectively. Seventy-eight (61.42%) patients exhibited a UIP pattern through HRCT. Patients showing RA-ILD progression received a higher mean initially prescribed daily prednisone dose (4.46 mg/day vs. 2.66 mg/day, *p* = 0.005), had a higher mean baseline DAS28-ESR (4.55 vs. 3.98, *p* = 0.025), and exhibited a lower mean predicted DLCO at the baseline (66.8% vs. 75.3%, *p* = 0.066 before imputation, *p* = 0.002 after imputation) than patients without progression.

### 3.2. Risk of RA-ILD Progression Between the csDMARD-Only and b/tsDMARD-Exposed Groups

Figure 1 shows the Kaplan–Meier survival curve illustrating the risk of ILD progression between the csDMARD-only and b/tsDMARD-exposed groups. As indicated by this curve, the csDMARD-only group generally exhibited a higher rate of ILD worsening than the b/tsDMARD-exposed group; however, this difference did not reach statistical significance (*p* = 0.32 via a log-rank test). The median survival time was 5.53 years in the csDMARD-only group and 7.31 years in the b/tsDMARD-exposed group.

### 3.3. Association Between the Risk of RA-ILD Progression and b/tsDMARDs Exposure

The association between b/tsDMARD exposure and ILD progression, accounting for changes in drug exposure over time, was evaluated using a time-varying Cox regression model. The AIC value of Cox regression was the smallest when stratified by the year of the first DMARD treatment initiation (2010s or 2020s) (AIC = 479.08), as compared to when it was excluded (AIC = 510.30) or when the variables were included as standard covariates (AIC = 511.78). Thus, the stratified Cox model was chosen for further analysis (Table 2). As a result of stepwise selection with minimal AIC (Model 1), significant covariates associated with RA-ILD progression included DAS28-ESR (adjusted HR [aHR]: 1.344, 95% CI: 1.136–1.590, *p* = 0.001) and initially prescribed prednisone dose (aHR: 1.078, 95% CI: 1.011–1.151, *p* = 0.023). The percentage of predicted DLCO at the baseline was significantly associated with the outcome in the univariate analysis (HR: 0.970, 95% CI: 0.952–0.988, *p* = 0.001); however, it was excluded in the multivariate analysis, owing to a lack of statistical significance after adjustment for other covariates After adjustment for all of these covariates, the estimated aHR for b/tsDMARD exposure was 0.937 (95% CI: 0.475–1.849, *p* = 0.851), indicating no significant association between b/tsDMARD exposure and RA-ILD progression.

### 3.4. Sensitivity Analysis

TNF inhibitors have been suggested in previous studies to increase the risk of ILD progression; in contrast, non-TNF b/tsDMARDs stabilize it [1,2,10,22,23]. To test the sensitivity of our findings, we excluded patients exposed to TNF inhibitors and re-analyzed the association between b/tsDMARD exposure and RA-ILD progression. Even after excluding the TNF inhibitor group, the Kaplan–Meier curve indicated a higher ILD progression rate in the csDMARD-only group; however, this difference remained statistically non-significant (*p* = 0.56 by log-rank test; Figure 2). The stratified time-varying Cox analysis revealed that the best model, as determined via the AIC, was the one adjusted for DAS28-ESR and initially prescribed prednisone dose (Table 3). The aHR for non-TNF b/tsDMARD exposure was 1.111 (95% CI: 0.561–2.204), with the direction of the association opposite to that of the previous analysis; nonetheless, this association remained statistically non-significant (*p* = 0.762).

### 3.5. Adverse Events and Mortality

No significant differences in adverse events or mortality were observed between patients with RA-ILD who were treated with csDMARDs and those who received b/tsDMARDs, irrespective of ILD progression status (Table 4). Among patients with progressed RA-ILD, hospitalization for any cause and infection rates were slightly higher in the csDMARD-only group than in the b/tsDMARD-exposed group (34.5% vs. 10.0%); however, these differences did not reach statistical significance. Death occurred exclusively in the csDMARD-only group, with a total of nine reported cases; nevertheless, no statistically significant difference was noted when compared to the b/tsDMARD-exposed group. The causes of death included ILD worsening (*n* = 3), bacterial pneumonia (*n* = 2), COVID-19 infection (*n* = 1), septic shock due to anastomosis site leakage (*n* = 1), and lung cancer (*n* = 2).

## 4. Discussion

This study provides key insights into the diagnosis and prognosis of RA-ILD, emphasizing the importance of comprehensive clinical evaluation and monitoring strategies. The findings demonstrate that PFTs, HRCT findings, and symptom changes are critical parameters for assessing ILD progression and prognosis. More than half of patients with RA-ILD experienced ILD progression during a mean follow-up period of 2.8 years, highlighting the need for early intervention and close monitoring.

A crucial insight from this study is that higher disease activity (DAS28-ESR) and a higher initial glucocorticoid dose were significantly associated with RA-ILD progression. However, exposure to b/tsDMARDs, including non-TNF b/tsDMARDs, was not significantly associated with RA-ILD progression, even after adjustment. Furthermore, no significant differences in adverse events or mortality were observed between patients treated with csDMARDs and those receiving b/tsDMARDs, irrespective of ILD progression status. This finding suggests that RA disease control itself may play a more central role in prognosis than the choice of specific DMARD therapy.

The recent ACR/American College of Chest Physicians guideline conditionally recommends mycophenolate, azathioprine, and rituximab as the first-line treatment for RA-ILD; in contrast, leflunomide, methotrexate, TNF inhibitors, and abatacept are not conditionally recommended as first-line treatment options for ILD [24]. However, the selection of treatment depends not only on pulmonary manifestations but also on extra-pulmonary factors, including inflammatory arthritis and patient-specific factors such as comorbidities. While mycophenolate is generally favored as a first-line treatment option for RA-ILD, immunosuppressive drugs employed against connective tissue disease-ILD and antifibrotic drugs are not effective against arthritis [24]. Indeed, not all patients with RA-ILD are treated with a primary focus on ILD. Treatment decisions are comprehensively made while considering clinical symptoms, RA disease activity, pulmonary function, HRCT findings, the presence of progression, patient age, and comorbidities that may increase the risk of adverse events [25]. In clinical practice, RA-ILD is often monitored while using DMARDs to manage arthritis, allowing clinicians to detect ILD development or progression. Therefore, further studies should be conducted to investigate the potential influence of DMARDs on RA-ILD progression. Our study found no significant difference in the impact of csDMARDs and b/tsDMARDs on RA-ILD progression, reaffirming the importance of controlling disease activity in the context of RA-ILD.

Accumulative evidence underscores the importance of achieving remission or at least low disease activity in arthritis to prevent the emergence, progression, or acute exacerbation of RA-ILD. In a recent U.S. prospective cohort study involving approximately 1400 patients with RA without baseline RA-ILD who were followed for 9 years on average, poorly controlled RA disease activity, as measured using the DAS28, was significantly associated with RA-ILD development [26]. Similar to our study, two retrospective single-center studies reported an association between higher disease activity based on the Clinical Disease Activity Index score and RA-ILD progression [5,6]. Furthermore, a retrospective case-control study demonstrated an association between uncontrolled arthritis activity during tocilizumab treatment and the acute exacerbation of RA-ILD [27]. Thus, achieving remission or low disease activity is crucial not only for managing RA itself but also for preventing RA-ILD development and progression.

Several studies have demonstrated the association of RA-ILD progression with older age, lower FVC or DLCO, male sex, smoking history, a greater extent of ILD in HRCT, and a UIP pattern in HRCT [5,6,28]. In our study, the percentage of predicted DLCO at the baseline was the only univariate predictor significantly associated with progression. Other variables such as older age and smoking history showed a similar trend toward being poor predictors of RA-ILD progression but did not reach statistical significance, which may be attributable to our sample size and the presence of missing data.

Non-TNF inhibitors such as abatacept, tocilizumab, and rituximab have been shown to significantly and independently reduce the risk of exacerbations and/or death related to RA-ILD [1,2,11,13]. However, the use of csDMARDs in RA-ILD remains controversial. The impact of methotrexate and leflunomide, which are recommended as first-line treatment for RA [29], on ILD requires further investigation. Recent research suggests that methotrexate use is not associated with an increased risk of ILD development or progression in patients with RA [2,8,24]. Leflunomide is rarely linked to the development or worsening of ILD; nonetheless, evidence supporting its direct beneficial effect on ILD outcomes remains limited [2,7,30]. Furthermore, while methotrexate is generally not strongly associated with ILD progression in observational studies, it can rarely induce idiosyncratic pneumonitis, highlighting the need for careful monitoring [2]. Our findings revealed that csDMARDs were not significantly associated with ILD progression compared to b/tsDMARDs, including non-TNF b/tsDMARDs. Moreover, we did not find evidence to support that b/tsDMARDs are more effective than csDMARDs in preventing RA-ILD progression. Our sensitivity analysis, conducted after four TNF inhibitor users were excluded from the b/tsDMARD-exposed group (*n* = 22), revealed that the risk of progression had an HR >1; however, it was not statistically significant. This analysis was performed based on previous studies suggesting that TNF inhibitors are associated with the progression of RA-ILD, whereas non-TNF b/tsDMARDs may have a stabilizing effect on ILD. Although the exclusion of TNF inhibitors resulted in an opposite trend, the small number of TNF inhibitor users necessitates a cautious interpretation of these findings. These results highlight the need for further research to clarify the impact of b/tsDMARDs, particularly non-TNF b/tsDMARDs, on RA-ILD progression [31].

Among patients in the b/tsDMARD-exposed group, 77.3% were also receiving concomitant csDMARDs. A retrospective national multicenter study reported similar effects of abatacept, whether used as monotherapy or in combination with csDMARDs, on lung function and radiographic changes in RA-ILD patients [32]. To date, no new definitive evidence has been established showing that b/tsDMARD monotherapy is superior to combination therapy in the treatment of RA. Therefore, b/tsDMARDs therapy in RA is recommended with csDMARD [29]. Further comparative studies are needed to evaluate the impact of b/tsDMARD monotherapy versus b/tsDMARDs combined with csDMARDs on the progression of RA-ILD.

This study involved several limitations. First, the present study was a retrospective, observational study in which the intervals for PFTs and chest CT scans were determined at the discretion of treating physicians. Additionally, a significant proportion of patients did not have available FVC or DLco data at the baseline, which may have limited the assessment of ILD progression. Despite the use of imputation methods to address this issue, further studies with more complete pulmonary function data are needed. Despite the use of a time-varying Cox model to address variability in the types and duration of drug exposure among patients, potential biases related to the timing of assessments remain. Second, while this study utilized both PFTs and chest CT scans to examine disease progression—an advantage over studies relying solely on PFTs—there may be variability in the interpretation of CT findings, owing to the reliance on assessments by radiologists from different institutions. Although expert thoracic radiologists evaluated ILD progression using HRCT, we did not assess inter-reader agreement, which could have influenced the consistency of the results. Future studies should aim to enhance the reliability of progression evaluation by integrating advanced tools such as artificial intelligence-based analysis [33]. This could standardize assessments and reduce variability, improving the overall robustness of findings. Third, this study primarily compared the effects of b/tsDMARDs and csDMARDs on RA-ILD progression. However, the number of patients in the b/tsDMARD group was significantly lower than that in the csDMARD group, which may have influenced the statistical power of the analysis. Indeed, the sample size for individual drugs within the non-TNF inhibitor b/tsDMARD category was insufficient to allow for comparisons at the level of specific agents. This limitation restricted our ability to draw conclusions about the potential differential effects of individual drugs on ILD progression. Future studies with larger cohorts should be conducted to evaluate the impact of specific therapeutic agents on RA-ILD outcomes. Lastly, one of the limitations of this study is the relatively short DMARD exposure period. Patients were included if they had been on DMARDs for at least three months, which may be insufficient to fully assess treatment efficacy. Among all 127 study participants, 49 patients (38.5%) had an exposure duration of less than 6 months on DMARDs. A longer exposure period could provide a more comprehensive evaluation of the impact of DMARDs on RA-ILD.

## 5. Conclusions

This study highlights that effective RA disease control is paramount in managing RA-ILD prognosis. While csDMARDs and b/tsDMARDs did not show significant differences in ILD progression, achieving low disease activity remains a crucial therapeutic goal. Our findings suggest that effective disease activity management may play a crucial role in slowing RA-ILD progression.

## Figures and Tables

**Figure 1 diagnostics-15-00800-f001:**
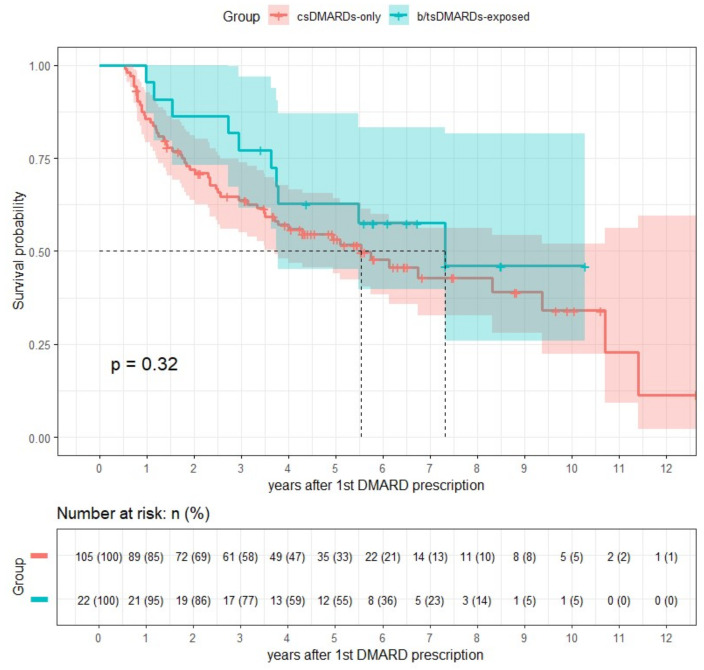
Kaplan–Meier curve for RA-ILD progression in the csDMARD-only group versus b/tsDMARD-exposed group. The dashed lines indicate the median survival times for each group.

**Figure 2 diagnostics-15-00800-f002:**
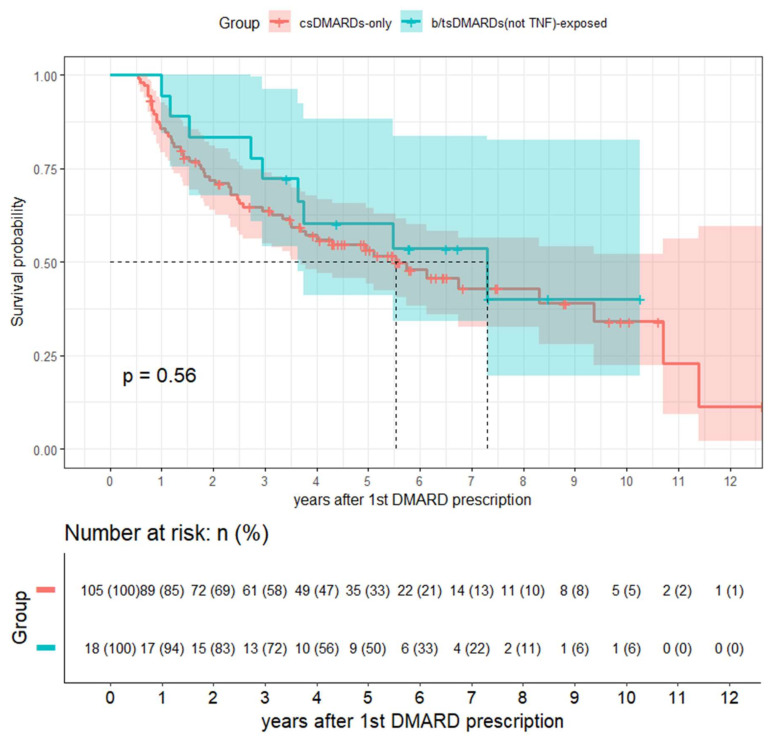
Kaplan–Meier curve for RA-ILD progression in the csDMARD-only group vs. b/tsDMARD-exposed group (excluding TNF inhibitors). The dashed lines indicate the median survival times for each group.

**Table 1 diagnostics-15-00800-t001:** Baseline and follow-up characteristics of patients with RA-ILD.

Variable	Total (*n* = 127)	Patients Without ILD Progression (*n* = 62)	Patients with ILD Progression (*n* = 65)	*p*-Value
Medication use during the observation period, *n* (%)				0.722
csDMARD-only	105 (82.68)	50 (80.65)	55 (84.62)	
b/tsDMARD-exposed	22 (17.32)	12 (19.35)	10 (15.38)	
Duration of all DMARDs (days)	1541.65 ± 1026.14	2048.65 ± 894.55	1058.05 ± 907.62	<0.001
Age at baseline (years)	63.01 ± 8.21	63.02 ± 8.43	63.00 ± 8.05	0.991
Female sex, *n* (%)	70 (55.12)	37 (59.68)	33 (50.77)	0.406
First medication year, *n* (%)				0.922
2010s	112 (88.19)	54 (87.10)	58 (89.23)	
2020s	15 (11.81)	8 (12.90)	7 (10.77)	
Current or ever-smoker, *n* (%)	45 (35.43)	22 (35.48)	23 (35.38)	1.000
Initially prescribed prednisone dose (mg/day)	3.58 ± 3.67	2.66 ± 3.38	4.46 ± 3.75	0.005
Prednisone use at follow-up, *n* (%)	78 (61.42)	33 (53.23%)	45 (69.23)	0.095
Rheumatoid factor positivity, *n* (%)	124 (99.20)	61 (100.00%)	63 (98.44)	1.000
Number of patients with missing values	2	1	1	
Anti-cyclic citrullinated peptide positivity, *n* (%)	106 (95.50)	50 (94.34)	56 (96.55)	0.668
Number of patients with missing values	16	9	7	
CRP at baseline (mg/dL)	1.86 ± 3.20	1.86 ± 3.49	1.86 ± 2.93	0.997
Imputed CRP (mg/dL)	1.86 ± 3.18	1.86 ± 3.49	1.86 ± 2.88	0.996
Number of patients with missing values	2	0	2	
ESR at baseline (mm/hour)	47.42 ± 33.16	44.31 ± 34.36	50.44 ± 31.92	0.302
Imputed ESR at baseline (mm/hour)	47.42 ± 33.02	44.31 ± 34.36	50.39 ± 31.67	0.302
Number of patients with missing values	1	0	1	
DAS28-ESR at baseline	4.26 ± 1.34	3.98 ± 1.27	4.55 ± 1.35	0.025
Imputed DAS28-ESR at baseline	4.27 ± 1.23	4.02 ± 1.20	4.50 ± 1.22	0.026
Number of patients with missing values	17	6	11	
Percent predicted FVC at baseline	89.17 ± 17.14	90.78 ± 15.04	88.40 ± 18.17	0.72
Imputed percent predicted FVC at baseline	90.35 ± 13.03	92.12 ± 09.90	88.65 ± 15.32	0.131
Number of patients with missing values	62	41	21	
Percent predicted DLCO at baseline	69.55 ± 16.91	75.31 ± 16.62	66.88 ± 16.56	0.066
Imputed percent predicted DLCO at baseline	73.12 ± 13.28	76.77 ± 11.19	69.64 ± 14.24	0.002
Number of patients with missing values	67	43	24	
Honeycombing pattern in HRCT at baseline, *n* (%)	70 (56.00)	35 (58.33)	35 (53.85)	0.746
Imputed honeycombing pattern in HRCT at baseline, *n* (%)	72 (56.69)	37 (59.68)	35 (53.85)	0.629
Number of patients with missing values	2	2	0	
HRCT pattern at baseline, *n* (%)				0.782
Usual interstitial pneumonia	78 (61.42)	37 (59.68)	41 (63.08)	
Nonspecific interstitial pneumonia	47 (37.01)	23 (37.10)	24 (36.92)	
Lymphocytic interstitial pneumonia	1 (0.79)	1 (1.61)	0 (0.00)	
Medication use during the observation period, *n* (%)				
csDMARDs				
Methotrexate	71 (55.91)	36 (58.06)	35 (53.85)	0.764
Leflunomide	15 (11.81)	8 (12.90)	7 (10.77)	0.922
Sulfasalazine	34 (26.77)	16 (25.81)	18 (27.69)	0.969
Hydroxychloroquine	70 (55.12)	31 (50.00)	39 (60.00)	0.340
Tacrolimus	60 (47.24)	29 (46.77)	31 (47.69)	1.000
b/tsDMARDs				
TNF inhibitors	4 (3.15)	3 (4.84)	1 (1.54)	0.357
Abatacept	13 (10.24)	7 (11.29)	6 (9.23)	0.928
Tocilizumab	7 (5.51)	3 (4.84)	4 (6.15)	1.000
Rituximab	1 (0.79)	1 (1.61)	0 (0.00)	0.488
Targeted synthetic DMARD	1 (0.79)	0 (0.00)	1 (1.54)	1.000

Data were reported as means ± standard deviations, unless otherwise stated. DMARD: disease-modifying antirheumatic drugs; csDMARD: conventional synthetic DMARD; b/tsDMARD: biologic/targeted synthetic DMARD; RA-ILD: rheumatoid arthritis-associated interstitial lung disease; CRP: C-reactive protein; ESR: erythrocyte sedimentation rate; DAS28-ESR: Disease Activity Score-28 with erythrocyte sedimentation rate; HRCT: high-resolution computed tomography; FVC: forced vital capacity; DLCO: diffusing capacity of the lungs for carbon monoxide; TNF: tumor necrosis factor.

**Table 2 diagnostics-15-00800-t002:** Risk factors for ILD progression in patients with RA-ILD stratified by initial DMARD prescription year (2010s vs. 2020s).

Variable	Crude HR * (95% CI)	*p*-Value	aHR ^†^ (95% CI)	*p*-Value
b/tsDMARD-exposed, yes	0.735 (0.372, 1.45)	0.374	0.937 (0.475, 1.849)	0.851
Sex, male	1.258 (0.773, 2.049)	0.356		
Age	1.022 (0.992, 1.053)	0.153		
Log-transformed CRP	1.091 (0.961, 1.238)	0.179		
ESR	1.009 (1.001, 1.016)	0.019		
DAS28-ESR	1.439 (1.178, 1.758)	4 × 10^−4^	1.344 (1.136, 1.590)	0.001
Initially prescribed prednisone dose	1.088 (1.029, 1.150)	0.003	1.078 (1.011, 1.151)	0.023
Honeycombing pattern in HRCT, yes	0.814 (0.494, 1.343)	0.421		
Smoking status, current or ever smoker	1.588 (0.804, 3.136)	0.183		
Percentage predicted FVC at baseline	0.996 (0.974, 1.018)	0.693		
Percentage predicted DLCO at baseline	0.970 (0.952, 0.988)	0.001		
AIC = 479.08				
Concordance index = 0.651 (se = 0.035)				

HR: hazard ratio; CI: confidence interval; aHR: adjusted hazard ratio; DMARD: disease-modifying antirheumatic drugs; b/tsDMARD: biologic/targeted synthetic DMARD; RA-ILD: rheumatoid arthritis-associated interstitial lung disease; CRP: C-reactive protein; ESR: erythrocyte sedimentation rate; DAS28-ESR: Disease Activity Score-28 with erythrocyte sedimentation rate; HRCT: high-resolution computed tomography; FVC: forced vital capacity; DLCO: diffusing capacity of the lungs for carbon monoxide. * Hazard ratios were estimated using a simple Cox proportional hazards model stratified by the year of the first DMARD prescription. The variable “b/tsDMARD-exposed, yes” indicates exposure during the follow-up period. ^†^ aHRs were estimated using a multiple time-varying Cox proportional hazards model stratified by the year of the first DMARD prescription. The variable “b/tsDMARD-exposed, yes” was treated as a time-varying covariate.

**Table 3 diagnostics-15-00800-t003:** Risk factors for ILD progression in patients with RA-ILD excluding TNF inhibitor-exposed cases stratified by initial DMARD prescription year (2010s vs. 2020s).

Variable	Crude HR * (95% CI)	*p*-Value	aHR ^†^ (95% CI)	*p*-Value
b/tsDMARD-exposed, yes	0.837 (0.412, 1.702)	0.624	1.111 (0.561, 2.204)	0.762
Sex, male	1.201 (0.735, 1.963)	0.465		
Age	1.020 (0.989, 1.052)	0.216		
Log-transformed CRP	1.082 (0.953, 1.227)	0.225		
ESR	1.008 (1.001, 1.015)	0.024		
DAS28-ESR	1.442 (1.179, 1.763)	4 × 10^−4^	1.327 (1.123, 1.568)	0.001
Initially prescribed prednisone dose	1.088 (1.029, 1.150)	0.003	1.081 (1.013, 1.152)	0.018
Honeycombing pattern in HRCT, yes	0.800 (0.482, 1.326)	0.387		
Smoking status, current or ever-smoker	1.717 (0.868, 3.399)	0.121		
Percentage predicted FVC at baseline	0.995 (0.973, 1.017)	0.640		
Percentage predicted DLCO at baseline	0.970 (0.952, 0.988)	0.001		
AIC = 465.62				
Concordance index = 0.655 (se = 0.034)				

HR: hazard ratio; CI: confidence interval; aHR: adjusted hazard ratio; DMARD: disease-modifying antirheumatic drugs; b/tsDMARD: biologic/targeted synthetic DMARD; RA-ILD: rheumatoid arthritis-associated interstitial lung disease; CRP: C-reactive protein; ESR: erythrocyte sedimentation rate; DAS28-ESR: Disease Activity Score-28 with erythrocyte sedimentation rate; HRCT: high-resolution computed tomography; FVC: forced vital capacity; DLCO: diffusing capacity of the lungs for carbon monoxide. * Hazard ratios were estimated using a simple Cox proportional hazards model stratified by the year of the first DMARD prescription. The variable “b/tsDMARD-exposed, yes” indicates exposure during the follow-up period. ^†^ aHRs were estimated using a multiple time-varying Cox proportional hazards model stratified by the year of the first DMARD prescription. The variable “b/tsDMARD-exposed, yes” was treated as a time-varying covariate.

**Table 4 diagnostics-15-00800-t004:** Adverse events and mortality according to DMARD use in patients with RA-ILD.

	Patients Without ILD Progression (*n* = 62)			Patients with ILD Progression (*n* = 65)		
	csDMARDs (*n* = 50)	b/tsDMARDs (*n* = 12)	*p*-Value	csDMARDs (*n* = 55)	b/tsDMARDs (*n* = 10)	*p*-Value
Hospitalization						
Any cause	10 (20.0)	3 (25.0)	0.703	19 (34.5)	1 (10.00%)	0.156
ILD worsening	5 (10.0)	1 (8.3)	1.000	6 (10.9)	0 (0.0)	0.579
Infection	3 (6.0)	1 (8.3)	1.000	11 (20.0)	0 (0.0)	0.190
Tuberculosis	0 (0.0)	0 (0.0)	1.000	2 (3.6)	0 (0.0)	1.000
Shingles	1 (2.0)	0 (0.0)	1.000	0 (0.0)	0 (0.0)	1.000
Malignancy	1 (2.0)	0 (0.0)	1.000	3 (5.4)	0 (0.0)	1.000
MACE	1 (2.0)	0 (0.0)	1.000	1 (1.8)	0 (0.0)	1.000
Mortality	3 (6.0)	0 (0.0)	1.000	6 (10.9)	0 (0.0)	0.579

Data were reported as *n* (%) for categorical variables. DMARD: disease-modifying antirheumatic drugs; csDMARD: conventional synthetic DMARD; b/tsDMARD: biologic/targeted synthetic DMARD; RA-ILD: rheumatoid arthritis-associated interstitial lung disease.

## Data Availability

The data supporting the findings of this study are available from the corresponding author upon reasonable request. Due to privacy and ethical restrictions, the data are not publicly available.

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
