# Peer review of "Impact of csDMARDs vs. b/tsDMARDs on the Prognosis of Rheumatoid Arthritis-Associated Interstitial Lung Disease: A Multicenter, Retrospective Study"

_diagnostics, 2025, doi:10.3390/diagnostics15070800_

Round 1
Reviewer 1 Report
Comments and Suggestions for Authors
Correct punctuation: punctuation before the bibliographic reference in the sentence should be removed.
Methods
Line 111: replace prescription with treatment.
Line 139: It is not specified from where patients' chest HRTCs are extrapolated and especially how the extent of ILD to HRTC is quantified. This data is important because it is one of the factors in defining whether ILD is stable or progressing. Therefore, it is requested that this information be added.
Line 143: From where are patients' clinical data extracted (data base?). Please specify
Line 154: Are all patients in the b/tsDMARD-exposed group also being treated with csDMARD? Please specify.
Results
Line 198: Line 117 states that 156 patients were recruited, while the results show that the patients were reduced to 127 without specifying the reason. Therefore, I think it is appropriate to delete the sentence in line 116-117 because it is unnecessary, misleading and because it is a result that should not be included in materials and methods anyway.
Table 1:
Prednisone dose is not mg/dl. Please correct.
Are the patients with HRTC pattern “others” as described in line 140 all with lymphocytic interstitial pneumonia (a very atypical fact, given the rarity of such a pattern in RA) or is it an omission? Justify this abnormality.
Why was it written “medication administered at follow-up?” Medication is an inclusion criterion and not a therapy prescribed at a follow-up visit. Please specify this information because it is not clear.
The table shows that half of the patients do not have FVC or DLco, which are critical for the assessment of progression. I think this is a fact that needs to be further investigated and emphasized.
In addition, it would be appropriate to describe in the results and may be even comment on it in the discussion on the treatment of the recruited patients, because it emerges that many patients were taking csDMARDs in combination. It is requested to specify this data and comment in the discussion. Also, were the patients on b/tsDMARDs also taking csDMARDs? this finding should be specified in the results and commented in the discussion.
Results
Line 218: The dosage of prednisone should be specified whether daily, annually, or weekly.
LIMITS
I would introduce the short limit of DMARD exposure: in fact, patients who had been taking DMARDs for only three months were recruited. The exposure time of the drug is short to evaluate its efficacy. It would have been optimal to increase this period to six months.
it would be appropriate to emphasize the low number of patients in b/tsDMARD compared with the group in csDMARD because the purpose of the article is to compare the progression of ILD in these two groups, but they have a very different sample so this may lead to results of low statistical quality.
Author Response
Comments 1: Correct punctuation: punctuation before the bibliographic reference in the sentence should be removed.
Response 1: Thank you for your careful review and valuable feedback. We have carefully revised the manuscript to remove any punctuation before the bibliographic references, ensuring consistency throughout the text.
Comments 2: Methods Line 111: replace prescription with treatment.
Response 2: Thank you for your suggestion. We have replaced "prescription" with "treatment" in Line 111 as requested.
Comments 3: Line 139: It is not specified from where patients' chest HRTCs are extrapolated and especially how the extent of ILD to HRTC is quantified. This data is important because it is one of the factors in defining whether ILD is stable or progressing. Therefore, it is requested– that this information be added.
Response 3: Thank you for your valuable comment. To clarify the source and quantification method of chest HRCT assessments, we have added the following statement:
"Regarding chest HRCT scans, the increased extent of fibrosis was qualitatively assessed by a board-certified thoracic radiologist at each participating center by comparing it with the baseline HRCT. The assessment was primarily based on the extent and phenotype of ILD."
Comments 4: Line 143: From where are patients' clinical data extracted (data base?). Please specify
Response 4: We have clarified the source of patients' clinical data by adding the following statement:
"Patients' clinical data, including the index date, were extracted from the hospital’s electronic medical records."
Comments 5: Line 154: Are all patients in the b/tsDMARD-exposed group also being treated with csDMARD? Please specify.
Response 5: Thank you for your insightful comment. To clarify the classification of patients in the b/tsDMARD-exposed group, we have added the following statement:
"If the patients started with csDMARDs and later added b/tsDMARDs, this was treated as a time-varying exposure, with the switch point defined as the time of bDMARD initiation. The observation period was considered until the last recorded b/tsDMARD prescription. If the patients started with b/tsDMARDs and later added csDMARDs, these patients were classified as b/tsDMARD users from baseline. The observation period was considered until the last recorded b/tsDMARD prescription."
Comments 6:Results Line 198: Line 117 states that 156 patients were recruited, while the results show that the patients were reduced to 127 without specifying the reason. Therefore, I think it is appropriate to delete the sentence in line 116-117 because it is unnecessary, misleading and because it is a result that should not be included in materials and methods anyway.
Response 6: As per your suggestion, we have deleted the sentence in Lines 116–117.
Comments 7: Table 1: Prednisone dose is not mg/dl. Please correct.
Response 7: We have corrected the prednisone dose unit from mg/dL to mg/day in Table 1.
Comments 8: Are the patients with HRTC pattern “others” as described in line 140 all with lymphocytic interstitial pneumonia (a very atypical fact, given the rarity of such a pattern in RA) or is it an omission? Justify this abnormality.
Response 8: Thank you for your insightful comment.
We have revised the manuscript to clarify the classification of ILD as follows:
"ILD was classified into nonspecific interstitial pneumonia, UIP including probable UIP, or other patterns including bronchiolitis obliterans, organizing pneumonia, lymphocytic interstitial pneumonitis, and mixed patterns."
As shown in Table 1, "Other patterns" were rare, and LIP was observed in only one case.
Comments 9: Why was it written “medication administered at follow-up?” Medication is an inclusion criterion and not a therapy prescribed at a follow-up visit. Please specify this information because it is not clear.
Response 9: To clarify, we have revised the phrase "medication administered at follow-up" to "medication use during the observation period" to better reflect that medication was an inclusion criterion rather than a therapy prescribed at a follow-up visit.
Comments 10: The table shows that half of the patients do not have FVC or DLco, which are critical for the assessment of progression. I think this is a fact that needs to be further investigated and emphasized.
Response 10: Thank you for your insightful comment. To address the high proportion of missing baseline FVC and baseline DLCO data, we applied the missForest algorithm for imputation before including these variables in the analysis (imputation performance: NRMSE = 0.230).
We acknowledge that this represents a limitation of our study. Accordingly, we have addressed this issue in the Discussion section with the following statement:
"Additionally, a significant proportion of patients did not have available FVC or DLCO data at baseline, which may limit the assessment of ILD progression. Despite the use of imputation methods to address this issue, further studies with more complete pulmonary function data are needed."
Comments 11: In addition, it would be appropriate to describe in the results and may be even comment on it in the discussion on the treatment of the recruited patients, because it emerges that many patients were taking csDMARDs in combination. It is requested to specify this data and comment in the discussion. Also, were the patients on b/tsDMARDs also taking csDMARDs? this finding should be specified in the results and commented in the discussion.
Response 11: Thank you for your insightful comment. To clarify the treatment patterns of the recruited patients, we have added the following details in the Results section:
"Among the patients exposed to b/tsDMARDs, 17 used both csDMARDs and b/tsDMARDs concurrently. Of these, 14 patients initially started with csDMARDs and later added b/tsDMARDs, while 3 patients initially started with b/tsDMARDs and later added csDMARDs. Among the concomitantly administered csDMARDs, MTX was the most commonly used, with 11 patients."
Additionally, we have expanded the Discussion section to comment on the implications of these findings:
"Among patients in the b/tsDMARD-exposed group, 77.3% were also receiving concomitant csDMARDs. A retrospective national multicenter study reported similar effects of abatacept, whether used as monotherapy or in combination with csDMARDs, on lung function and radiographic changes in RA-ILD patients [reference]. To date, no definitive evidence has been established that b/tsDMARD monotherapy is superior to combination therapy in the treatment of RA. Therefore, b/tsDMARD therapy in RA is recommended with csDMARDs [29]. Further comparative studies are needed to evaluate the impact of b/tsDMARD monotherapy versus b/tsDMARDs combined with csDMARDs on the progression of RA-ILD."
These revisions address the reviewer's concern by explicitly specifying the proportion of patients receiving combination therapy and discussing its relevance.
Comments 12: Results
Line 218: The dosage of prednisone should be specified whether daily, annually, or weekly.
Response 12: We have clarified the prednisone dosage specification by revising the sentence as follows:
Patients showing RA-ILD progression received a higher mean initially prescribed daily prednisone dose (4.46 mg/day vs. 2.66 mg/day, P=0.005),
Comments 13: LIMITS
I would introduce the short limit of DMARD exposure: in fact, patients who had been taking DMARDs for only three months were recruited. The exposure time of the drug is short to evaluate its efficacy. It would have been optimal to increase this period to six months.
Response 13: We acknowledge that the relatively short DMARD exposure period may be a limitation in assessing treatment efficacy. To address this, we have revised the manuscript to include the following statement in the Discussion section:
"Lastly, one of the limitations of this study is the relatively short DMARD exposure period. Patients were included if they had been on DMARDs for at least three months, which may be insufficient to fully assess treatment efficacy. Among all 127 study participants, 49 patients (38.5%) had an exposure duration of less than 6 months on DMARDs. A longer exposure period could provide a more comprehensive evaluation of the impact of DMARDs on RA-ILD."
This revision explicitly acknowledges the limitation and the need for further studies with a longer observation period.
Comments 14: It would be appropriate to emphasize the low number of patients in b/tsDMARD compared with the group in csDMARD because the purpose of the article is to compare the progression of ILD in these two groups, but they have a very different sample so this may lead to results of low statistical quality.
Response 14: To address this limitation, we have revised the Discussion section to include the following statement:
"Third, this study primarily compared the effects of b/tsDMARDs and csDMARDs on RA-ILD progression. However, the number of patients in the b/tsDMARD group was significantly lower than that in the csDMARD group, which may have influenced the statistical power of the analysis."
Reviewer 2 Report
Comments and Suggestions for Authors
The authors report a retrospective observational study that included 156 patients with RA-ILD at 13 referral hospitals in Sout Korea. They compared those receiving conventional synthetic disease-modifying antirheumatic drugs (csDMARD) with biologic/targeted synthetic disease-modifying antirheumatic drugs (b/tsDAMRDs). There was no statistical difference between the groups. Higher baseline disease activity score with erythrocyte sedimentation and initially prescribed prednisone dose were prognostic factors for ILD progression.
Specific Comments:
1) It would be helpful to be more explicit about how exactly other variables were derived (i.e. log-transformed CRP, reference for predicted FVC and DLCO, smoking status-current or ever smoker, progression on HRCT, worsening respiratory symptoms).
2) In table 1, you report the initially prescribed dose of prednisone in mg/dL.Clarify the dose unit.
3) In table 1, authors report that only 35 patients had honeycombing pattern on HRCT at baseline, however, they report 41 patients with Usual Interstitial Pneumonia on HRCT at baseline. How do you explain UIP pattern in more patients than honeycombing, which is needed for radiographic diagnosis of UIP?
4) In the discussion add the proportion of patients that are missing by excluding those on TNF inhibitor group in the sensitivity analysis, and how this could be relevant to your results
Author Response
Comments 1 : It would be helpful to be more explicit about how exactly other variables were derived (i.e. log-transformed CRP, reference for predicted FVC and DLCO, smoking status-current or ever smoker, progression on HRCT, worsening respiratory symptoms).
Response 1: Thank you for your insightful comment. To provide more explicit details on how variables were derived, we have revised the manuscript to include the following clarifications:
- Chest HRCT scans: The increased extent of fibrosis was qualitatively assessed by a board-certified thoracic radiologist at each participating center by comparing it with the baseline HRCT. The assessment was primarily based on the extent and phenotype of ILD. ILD was classified into nonspecific interstitial pneumonia, UIP (including probable UIP), or other patterns, including bronchiolitis obliterans, organizing pneumonia, lymphocytic interstitial pneumonitis, and mixed patterns. The presence of a honeycombing pattern on HRCT was also recorded [18].
- Predicted values for FVC and DLCO: These were derived using Korean reference standards [19, 20].
- Worsening respiratory symptoms: Defined as either worsening dyspnea or the onset of a new cough. Worsening dyspnea was assessed using the modified Medical Research Council (mMRC) scale (0–4) and was defined as an increase of at least one point.
- Smoking status: "Ever smokers" were defined as individuals who reported smoking at least 100 cigarettes in their lifetime, while "current smokers" were those who met this criterion and continued smoking.
- CRP log transformation: CRP values were log-transformed to achieve a normal distribution before statistical analysis. The original CRP data exhibited a skewness of 2.93, indicating substantial deviation from normality (skewness >1). To mitigate this skewness and approximate a normal distribution, a natural logarithm transformation was applied. After log transformation, the skewness was reduced to -0.12, ensuring a more symmetric distribution suitable for parametric analysis.
These additions ensure that the derivation of key variables is explicitly stated in the manuscript, addressing the reviewer’s concern.
Comments 2: In table 1, you report the initially prescribed dose of prednisone in mg/dL.Clarify the dose unit.
Response 2: We have corrected the unit of the initially prescribed prednisone dose from mg/dL to mg/day in Table 1 to ensure accuracy.
Comments 3: In table 1, authors report that only 35 patients had honeycombing pattern on HRCT at baseline, however, they report 41 patients with Usual Interstitial Pneumonia on HRCT at baseline. How do you explain UIP pattern in more patients than honeycombing, which is needed for radiographic diagnosis of UIP?
Response 3: Thank you for your insightful comment. To clarify, we revised the classification of ILD to include probable UIP, as reflected in the updated manuscript:
"ILD was classified into nonspecific interstitial pneumonia, UIP (including probable UIP), or other patterns, including bronchiolitis obliterans, organizing pneumonia, lymphocytic interstitial pneumonitis, and mixed patterns."
The discrepancy between the number of patients with honeycombing (n=35) and UIP pattern (n=41) arises because probable UIP cases were included in the UIP category. As per its definition,
"Probable UIP was defined as peripheral and basilar predominant pulmonary fibrosis with reticulation, little or no honeycombing, and absence of features to suggest another specific diagnosis."
Since honeycombing is not required for a probable UIP diagnosis, it is expected that the number of UIP cases exceeds the number of honeycombing cases.
Comments 4: In the discussion add the proportion of patients that are missing by excluding those on TNF inhibitor group in the sensitivity analysis, and how this could be relevant to your results
Response 4: Thank you for your valuable comment. In response to your suggestion, we have added the proportion of patients excluded from the TNF inhibitor group in the sensitivity analysis and discussed its potential relevance to our results. The following statement has been incorporated into the Discussion section:
"Our sensitivity analysis, conducted after excluding four TNF inhibitor users from the b/tsDMARD-exposed group (n=22), revealed that the risk of progression had an HR >1; however, it was not statistically significant. This analysis was performed based on previous studies suggesting that TNF inhibitors are associated with the progression of RA-ILD, whereas non-TNF b/tsDMARDs may have a stabilizing effect on ILD. Although the exclusion of TNF inhibitors resulted in an opposite trend, the small number of TNF inhibitor users necessitates cautious interpretation of these findings. These results highlight the need for further research to clarify the impact of b/tsDMARDs, particularly non-TNF b/tsDMARDs, on RA-ILD progression [31]."
Round 2
Reviewer 1 Report
Comments and Suggestions for Authors
The changes made are appropriate.
Author Response
Thank you for your positive feedback. We appreciate your careful review and are glad that the changes we made were appropriate.
Reviewer 2 Report
Comments and Suggestions for Authors
The authors report a retrospective observational study that included 156 patients with RA-ILD at 13 referral hospitals in Sout Korea. They compared those receiving conventional synthetic disease-modifying antirheumatic drugs (csDMARD) with biologic/targeted synthetic disease-modifying antirheumatic drugs (b/tsDAMRDs). There was no statistical difference between the groups. Higher baseline disease activity score with erythrocyte sedimentation and initially prescribed prednisone dose were prognostic factors for ILD progression.
Specific Comments:
1) It would be helpful to be more explicit about how exactly other variables were derived (i.e. log-transformed CRP, reference for predicted FVC and DLCO, smoking status-current or ever smoker, progression on HRCT, worsening respiratory symptoms).
2) In table 1, you report the initially prescribed dose of prednisone in mg/dL.Clarify the dose unit.
3) In table 1, authors report that only 35 patients had honeycombing pattern on HRCT at baseline, however, they report 41 patients with Usual Interstitial Pneumonia on HRCT at baseline. How do you explain UIP pattern in more patients than honeycombing, which is needed for radiographic diagnosis of UIP?
4) In the discussion, please add the proportion of patients that are missing by excluding those on TNF inhibitor group in the sensitivity analysis, and how this could be relevant to your results
Author Response

(The authors gave the same response as above.)
